# Frameshift variant in *MITF* gene in a large family with Waardenburg syndrome type II and a co-segregation of a *C2orf74* variant

Maan Abdullah Albarry[1], Muhammad Latif[2], Ahdab Qasem Alreheli[1], Mohammed A. Awadh[3], Ahmad M. Almatrafi[4], Alia M. Albalawi[2,5], Sulman Basit[2]*

1 Department of Ophthalmology, College of Medicine, Taibah University, Almadinah, Saudi Arabia, 2 Center for Genetics and Inherited Diseases, Taibah University, Almadinah, Saudi Arabia, 3 College of Applied Medical Sciences, Taibah University, Almadinah, Saudi Arabia, 4 Department of Biology, College of Science, Taibah University, Almadinah, Saudi Arabia, 5 Department of Biology, College of Science, King Abdulaziz University, Jeddah, Saudi Arabia

* sbasit.phd@gmail.com

**Data Availability Statement:** Raw reads (fastq files) have been uploaded to sequence read archives (SRA) of NCBI. The project details can be

## Abstract

Waardenburg syndrome (WS) is a hereditary disorder affecting the auditory system and pigmentation of hair, eyes, and skin. Different variants of the disease exist with the involvement of mutation in six genes. The aim of the study is to identify the genetic defects underlying Waardenburg syndrome in a large family with multiple affected individuals. Here, in this study, we recruited a large family with eleven affected individuals segregating WS type 2. We performed whole genome SNP genotyping, whole exome sequencing and segregation analysis using Sanger approach. Whole genome SNP genotyping, whole exome sequencing followed by Sanger validation of variants of interest identified a novel single nucleotide deletion mutation (c.965delA) in the *MITF* gene. Moreover, a rare heterozygous, missense damaging variant (c.101T>G; p.Val34Gly) in the *C2orf74* has also been identified. The C2orf74 is an uncharacterized gene present in the linked region detected by DominantMapper. Variants in *MITF* and *C2orf74* follows autosomal dominant segregation with the phenotype, however, the variant in *C2orf74* is incompletely penetrant. We proposed a digenic inheritance of variants as an underlying cause of WS2 in this family.

## Introduction

Waardenburg syndrome (WS) is a group of rare hereditary disorders. It is characterized by pigmentary defects of hair (white forelock), eyes (heterochromia iridis) and skin (hypo-pigmented skin), abnormalities in the inner ear (bilateral sensorineural hearing impairment), and dystopia canthorum (lateral displacement of the inner canthi of the eyes). WS syndrome has been categorized into four major types (WS1, WS2, WS3 and WS4). Further subtypes do exist though additional clinical manifestations are required for differential diagnosis of subtypes. WS1 (OMIM: 193500) and WS3 (OMIM: 148820) are characterized by the occurrence of dystopia canthorum. Absence of dystopia canthorum and other midface defects are the

accessed using the BioProject accession number PRJNA687138. Link for data accession is https://www.ncbi.nlm.nih.gov/Traces/study/?acc=PRJNA687138.

**Funding:** Ahmad M Almatrafi obtained funds under the research group grant number 10057 from the Deanship of Scientific Research (DSR), Taibah University Almadinah. The funder had no role in study design, data collection and analysis, decision to publish, or preparation of the manuscript. The funds were used for purchase of reagents and chemicals.

**Competing interests:** The authors have declared that no competing interests exist.

distinguishing features for WS2 (OMIM: 193510). Presence of dystopia canthorum and additional musculoskeletal anomalies of upper limbs demonstrates WS3 (OMIM: 148820) disease that is also known as Klein-Waardenburg syndrome. Individuals with WS4 have clinical manifestation of chronic intestinal pseudo-obstruction. WS4 is referred as Shah-Waardenburg syndrome or Waardenburg-Hirschsprung disorder (OMIM: 277580) [1–3]. Clinically WS1 and WS2 subtypes are the most common syndromes.

The clinical features of WS are not completely penetrant and highly varied expression has been observed [4]. This makes the differential diagnosis challenging [5]. Genetic diagnosis can help in identifying the type of WS in an individual patient. Mutations in at least six genes have been reported as an underlying cause of WS. Genotype-phenotype correlation between a causative gene mutation and various clinical features in WS are largely unclear. Until recently, mutations in *EDN3* (20q13.32), *EDNRB* (13q22.3), *MITF* (3p14p13), *PAX3* (2q36.1), *SNAI2* (8q11.21), and *SOX10* (22q13.1) have been reported in isolated patients as well as in families segregating WS [6–8]. For an instance, mutations of *PAX3* gene is recognized as a key indicator for the clinical features of WS1 and WS3, and mutations in *MITF*, *SOX10*, and *SNAI2* genes are identified in WS2. Moreover, genetic mutations in *EDN3*, *SOX10*, and *EDNRB* have been implicated in WS4 [9–14]. The majority of WS1 cases and some moderate WS3 cases are due to heterozygous mutations in the *PAX3* gene. Heterozygous mutations in *MITF* and *SOX10* are estimated to account for 30% of WS2 cases, and only 5% of the cases are attributed to the heterozygous mutations in *EDNRB* and *EDN3*, and homozygous mutations in *SNAI2* gene. In WS4, approximately 50% of cases have heterozygous mutations in *SOX10* and homozygous or heterozygous mutations in *EDNRB* and *EDN3* genes lead to WS4 in 20–30% of patients [15]. WS2 is phenotypically well characterized. It is further sub-divided into at least five types, WS2A (OMIM: 600193), WS2B (OMIM: 606662), WS2C (OMIM: 193510), WS2D (OMIM: 611584), and WS2E (OMIM: 608890).

We pursued our previous findings of identification of common genomic regions in multiple WS2 patients [16] and performed whole exome sequencing in two DNA samples of individuals from an extended family. We identified pathogenic variants in *MITF* and *C2orf74* genes as an underlying cause of WS2 phenotype.

## Materials and methods

### Ethical approval and sample collection

This study was approved by the institutional ethics review committee at the College of Medicine, Taibah University (study approval ID 051-02-2017) and the collaborating hospital. Written informed consents were obtained from all individuals included in the study or from their legal guardians. All experiments on human DNA were carried out in accordance with the declaration of Helsinki. A large six generation pedigree was drawn by interviewing the elders of the family (Fig 1). Blood samples were collected from nineteen individuals (IV:1, IV:2, IV:3, IV:4, IV:5, IV:6, IV:7, V:1, V:2, V:3, V:4, V:5, V:6, V:7, V:8, V:9, VI:1, VI:2, VI:3) including eleven affected individuals (IV:2, IV:3, IV:5, IV:7, V:2, V:4, V:6, V:7, V:8, VI:1, VI:2) in an EDTA containing vacutainers.

All subjects were examined at Magrabi Eye and Ear Hospital Almadinah Almunawwarah, Saudi Arabia. Detailed clinical history was recorded. During clinical phenotyping, colour of skin, hair and iris were noted. Moreover, special attention was given to observe dystopia canthorum and other anomalies such as medial eyebrow flare, limb defects, and Hirschsprung disease.

### Extraction of DNA and genetic analysis

Genomic DNA was extracted from the peripheral blood samples of all nineteen individuals using the QIAamp DNA Blood Mini Kit (QIAGEN GmbH—Germany). DNA was quantified

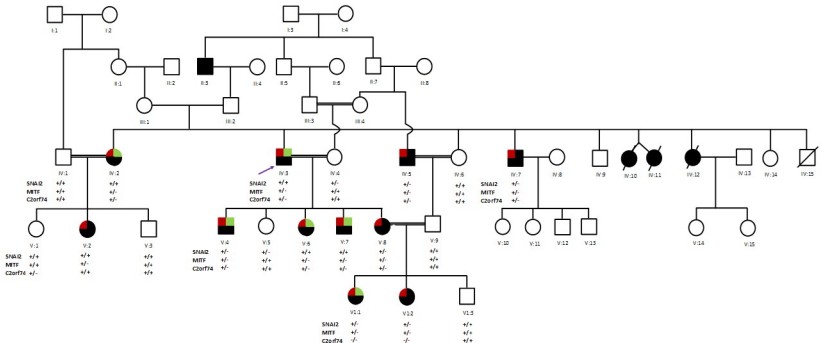

**Fig 1. An extended six generation pedigree chart of a Saudi family segregating Waardenburg syndrome type 2 in an autosomal dominant pattern.** Clear symbols represent unaffected individuals, whereas filled symbols represent affected individuals. Double lines are indicative of consanguineous unions. The index patient (proband) IV-3 is indicated by an arrow. Red and green colors shows deafness and iris atrophy/ heterochromia, respectively. Variants in SNAI2, MITF and C2orf74 have been shown beneath each individual. +/+ indicate wild type, +/- indicate heterozygous, and -/- indicate homozygous variant.

using micro-volume spectrophotometer (MaestroGen, Hsinchu City 30091, Taiwan). Whole genome SNP genotyping was performed as described elsewhere [17, 18] to identify the shared chromosomal segments in affected individuals. DNA from two affected individuals (VI:1, VI:2) were used to sequence the complete coding region of the human genome (whole exome sequencing). Whole exome was captured by Agilent SureSelect Target Enrichment Kit (v6) and sequenced by Macrogen Inc., using Illumina NovaSeq 6000 sequencing instrument (Illumina Inc, California, US). The length of the paired end reads were 150 bps, and the average coverage of the captured region was approximately 100x. Reads were aligned to the human reference genome (GRCh38/hg38). Genome Analysis Toolkit (GATK v3.7) was used to call the variants and variants were annotated using Illumina VariantStudio software. EVS (Exome Variant Server evsClient-v.0.0.16), GnomAD (Genome Aggregation Database v2.2.1), dbSNP (database of Single Nucleotide Polymorphisms dbSNP 2.0 Build 153), ExAC (Exome Aggregation Consortium), PolyPhen2 (Polymorphism Phenotyping v2), SIFT (Sorting Intolerant from Tolerant) and ClinVar were used to annotate, filter and prioritize the variants. Variants of interest were Sanger sequenced in both affected individuals by designing variant- specific primers. Primer3 tool was used for primer designing. Validated variants were screened in the DNA of all available individuals followed by segregation analysis.

## Results

### Clinical description of cases

Ocular hypopigmentation was present in all affected individuals except IV:5, IV:7, V:2, V:8, VI:2. All affected individuals have profound to mild hearing impairment (HI). W index was less than 1.95 mm for each affected individual, therefore, dystopia canthorum was considered as absent. This feature helped us in excluding WS type 1 (Fig 2). Moreover, no integumentary hypopigmentation in the form of white forelock or leukoderma was observed in any of the individual. Furthermore, synophrys, broad nasal root, hypoplasia of alae nasi or premature greying of the hair were not observed. Detailed examination ruled out the presence of upper limb malformations and chronic constipation. This ruled out WS type III and WS type IV in

**Fig 2. Clinical features of affected individuals with Waardenburg syndrome type II.** (A, B) A 16 years old son (V:4) of a proband with severe iris hypopigmentation OS together with a generalized hypopigmented fundus OS. (C, D, E) A 28 years old female daughter (V:6) of proband had sectoral iris atrophy and hypopigmentation OD together with a normally pigmented fundus OD. Her left eye showed severe iris hypopigmentation and atrophy associated with a severe generalized hypopigmented fundus. (F) A 25 years old niece (IV:5) showed normal ocular and fundal examinations, however, a large angle esotropia OS secondary to dense amblyopia was observed. (G, H) Audiological test assessment of a proband (IV:3) and his son (V:7), respectively, revealed profound bilateral sensorineural hearing loss. Figure has been taken from our previously published work [16].

this family. Based on the clinical phenotyping, the individuals in this study segregates WS type II in an autosomal dominant manner. Detail clinical analysis of each individual can be found elsewhere [16] (Fig 2).

## Genetic analysis identified a novel deletion mutation in the *MITF* gene and a missense variant in the *C2orf74* gene

Whole exome sequencing (WES) of two affected sisters (VI:1 and VI:2) produced ~62 million raw reads, with ~60 million high-quality reads aligned to the GRCh38/hg38 reference genome. Overall, a total of 101,648 high-quality variants were identified and 95.5% of these variants are present in the dbSNP142. Coding missense and indel variants and variants in the flanking intronic regions (within 10 bps) accounted for 48,934 variants, while 12,733 variants were synonymous. Various filter were applied to search for potential candidate variants including depth of coverage (DP >10), genotype quality (GQ >20), and unreported SNP. Only variants with low allelic frequency (below 1% in 1000G, ExAC) were taken into consideration. These settings yielded 67 rare coding variants (S1 Table). Variants in all known WS candidate genes (*EDN3*, *EDNRB*, *MITF*, PAX3, *SOX10*, *SNAI2*, and *TYRO3*) were searched and a novel rare heterozygous deletion mutation (c.965delA; p.Asn322fs) was identified in the *MITF* gene in both patients. Moreover, heterozygous missense variants in *SNAI3* (c.607C>T; p.Arg203Cys) and *TYRO3* (c.1037T>A; p.Ile346Asn) gene was identified in the exome data of both patients. Variant in *SNAI3* (c.607C>T; p.Arg203Cys) gene is rare in population and is probably damaging and deleterious as predicted by PolyPhen2 and SIFT, respectively. Variant in *TYRO3* (c.1037T>A; p.Ile346Asn) gene is present in population databases with high frequency (0.22 MAF) and is benign and tolerated as predicted by PolyPhen2 and SIFT, respectively.

Studies have shown that WNT pathway genes including *LEF-1* may modulate the WS2 phenotype in WS2 cases with *MITF* mutation [19, 20]. Therefore, exome data was searched for variants in WNT pathway genes (*LEF-1*, *RNF43*, *APC*, *ZNRF3*, *LRP4*, *LRP5*, *LRP6*, *ROR1*, *ROR2*, *GSK3*, *CK1*, *APC*, *BCL9*, and *BCL9L*) as well. No potentially pathogenic rare variant was identified. In order to identify variant(s) in other genes which might influence the expressivity of WS phenotype in our cases, exome data was filtered by using an unbiased and hypothesis-free approach. A rare missense variant (c.101T>G; p.Val34Gly) in the *C2orf74* gene was identified in both affected individuals. This variant is possibly damaging and deleterious as

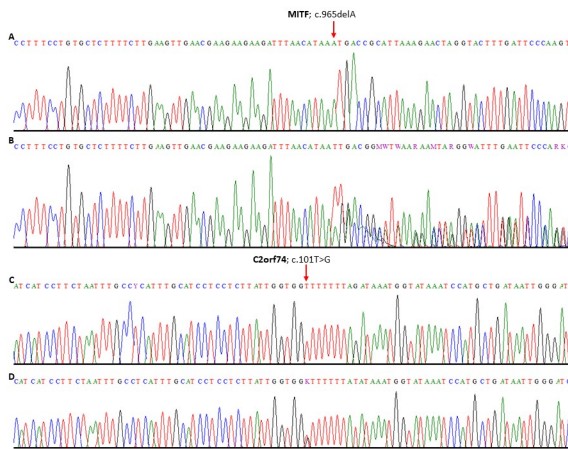

**Fig 3.** Electrophoretogram showing the partial reference sequence of *MITF* gene in the upper panel (a) while the panel b depicts a single base pair deletion in the *MITF* gene in affected individuals (b). Panel c shows the partial reference sequence of *C2orf74* gene and the lower panel shows sequence with heterozygous missense variant in the affected individuals (d).

predicted by PolyPhen2 and SIFT, respectively. Our previous genetic work on this family identified regions on chromosome 2p16.3-p15 and 18q21.33-q22.1 segregating with WS2 phenotype [16]. Interestingly, *C2orf 74* gene lies in the region on chromosome 2.

## Segregation analysis confirmed that the variants in *MITF* and *C2orf74* segregate with the disease phenotype in the family

The relevant coding exon of four genes (*C2orf74*, *MITF*, *SNAI3*, and *TYRO3*) were sequenced in all nineteen available members of a family including 11 affected individuals. Sanger sequencing validates the exome discovered mutations in both affected individuals and confirmed the segregation of *MITF* mutation in all family members as per autosomal dominant inheritance. Moreover, variant in *C2orf74* segregates with the phenotype in all family member except V:1 and V:2. V:1 is an unaffected individual carrying heterozygous variant while V:2 is an affected individual with wild type sequence (Fig 3). Variants in *SNAI2* and *TYRO3* are not segregating in the family (Fig 1).

## Discussion

Waardenburg syndrome (WS) is clinically characterized by occurrence of hearing impairment (HI) along with pigmentation abnormalities, including patchy hypopigmentation of the skin and hair and heterochromia iridis. In a subset of patients some other clinical features like Hirschsprung disease, dystopia canthorum, limb defects or neurological abnormalities are present. These features are often used to classify WS in four types (WS1, WS2, WS3, and WS4). WS is generally inherited in an autosomal dominant manner with variable penetrance. Inter- and intrafamilial variability in the expression of symptoms have also been observed. In some cases of WS, biallelic mutations have also been reported suggesting an autosomal recessive inheritance [8, 21–27].

Patients with Waardenburg syndrome type 2 (WS2) shows congenital sensorineural HI with depigmentation of hair, skin, and eyes. Dystopia canthorum is absent in WS2. It is a highly heterogeneous syndrome in terms of the underlying genetics. WS2 can be subdivided into five different types including WS2A (193510) caused by mutations in the *MITF* gene [4, 8,

12, 13, 20, 27–37], WS2B (600193) mapped to chromosome 1p21-p13.3, WS2C (606662) mapped to chromosome 8p23, WS2D (608890) caused by mutation in the *SNAI2* gene [38], and WS2E caused by mutations in the *SOX10* gene [39–46]. It is thought that haploinsufficiency is the underlying causative mechanism for WS2 [47]. Although, mutations in three genes (*MITF*, *SOX10*, and *SNAI2*) have been shown to cause WS2 subtypes, a number of cases remain unexplained at the molecular level. Moreover, inter- and intrafamilial phenotypic variability cannot be explained due to mutations in a single gene. Digenic inheritance or modulation of phenotype by modifier genes could possibly explain the variability and expressivity. Therefore, we embarked a genetic study using a large family segregating WS2. SNP genotyping followed by detection of shared regions identified two regions on chromosome 2 and 18 [16]. In this study, we sequenced complete exome in two affected individuals and identified candidate variants in *MITF* (c.965delA), *SNAI2* (c.607C>T) and *C2orf74* (c.101T>G) genes. Variant in *SNAI2* is not segregating with the disease phenotype therefore it was excluded as an underlying cause of WS2 in the family. *MITF* variant is perfectly segregating in all the affected members (Fig 3). Segregation analysis of *C2orf74* variant revealed that the variant segregates with the WS2 phenotype in all family members except V:1 and V:2. V:1 is an unaffected individual carrying heterozygous variant while V:2 is an affected individual with wild type sequence (Fig 1). Presence of heterozygous variant in an unaffected individual (V:1) might be due to incomplete expression of the phenotype.

Mutations in melanocyte inducing transcription factor (MITF), coding for a basic helix-loop-helix (BHLH) leucine zipper protein, are known to cause the WS2 phenotype due to defects in survival, proliferation, and migration of melanocytes [13, 28]. The deletion mutation (c.965delA) identified in this study lies in the BHLH domain and predicted to cause frameshift (p.Asn322fs) and stop codon seven amino acids downstream (Asn322Metfs*7). The missense variant (c.101T>G) in the *C2orf74* gene changes the conserved amino acid Valine to Glycine (p.Val34Gly). *C2orf74* is an uncharacterized gene and no functional data is available, however, Expression Atlas detected the expression of the gene in the eye (https://www.ebi.ac.uk/gxa/home). The gene *C2orf74* is present in the genomic region shared by all affected individuals [16] and therefore, we consider this as a candidate gene for WS2 phenotype. *C2orf74* gene might interact with MITF gene product and give rise to the spectrum of phenotype varying from severe phenotype with complete penetrance to partial features.

## Conclusion

In this study, we analysed a large family segregating Waardenburg syndrome type 2 to identify the underlying genetic defects. Whole genome SNP genotyping, whole exome sequencing and segregation analysis using Sanger approach was performed and a novel single nucleotide deletion mutation (c.965delA) in the *MITF* gene and a rare heterozygous, missense damaging variant (c.101T>G; p.Val34Gly) in the *C2orf74* was identified. Both variants follow autosomal dominant segregation with the phenotype, however, the variant in *C2orf74* is not completely penetrant. We proposed a digenic inheritance of variants as an underlying cause of WS2 in this family.

## Supporting information

**S1 Table.**
(XLSX)

## Acknowledgments

We are thankful to the patients and their families included in this research.

## Author Contributions

**Conceptualization:** Maan Abdullah Albarry, Sulman Basit.

**Data curation:** Muhammad Latif, Ahdab Qasem Alreheli, Mohammed A. Awadh, Ahmad M. Almatrafi.

**Formal analysis:** Muhammad Latif, Ahdab Qasem Alreheli, Ahmad M. Almatrafi.

**Funding acquisition:** Ahmad M. Almatrafi.

**Investigation:** Maan Abdullah Albarry, Ahdab Qasem Alreheli.

**Methodology:** Muhammad Latif, Ahdab Qasem Alreheli, Mohammed A. Awadh, Ahmad M. Almatrafi, Alia M. Albalawi, Sulman Basit.

**Supervision:** Sulman Basit.

**Validation:** Mohammed A. Awadh, Alia M. Albalawi.

**Visualization:** Maan Abdullah Albarry.

**Writing – original draft:** Sulman Basit.

**Writing – review & editing:** Alia M. Albalawi.

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
