## [Decision Letter · Decision Letter 0]

7 Dec 2020

PONE-D-20-28113

Frameshift variant in MITF gene in a large family with Waardenburg Syndrome type II and a Co-segregation of a C2orf74 variant

PLOS ONE

Dear Dr. Basit,

Thank you for submitting your manuscript to PLOS ONE. After careful consideration, we feel that it has merit but does not fully meet PLOS ONE’s publication criteria as it currently stands. Therefore, we invite you to submit a revised version of the manuscript that addresses the points raised during the review process.

Please provide the detailed explanation of the data analysis and also make the data available somewhere in the database archives.

We look forward to receiving your revised manuscript.

Kind regards,

Obul Reddy Bandapalli, MSc, PhD

Academic Editor

PLOS ONE

Journal Requirements:

2.We note that you have indicated that data from this study are available upon request. PLOS only allows data to be available upon request if there are legal or ethical restrictions on sharing data publicly. For more information on unacceptable data access restrictions, please see http://journals.plos.org/plosone/s/data-availability#loc-unacceptable-data-access-restrictions.

Reviewers' comments:

Reviewer's Responses to Questions

**Comments to the Author**

1. Is the manuscript technically sound, and do the data support the conclusions?

Reviewer #1: Yes

Reviewer #2: Partly

2. Has the statistical analysis been performed appropriately and rigorously? 

Reviewer #1: Yes

Reviewer #2: Yes

3. Have the authors made all data underlying the findings in their manuscript fully available?

Reviewer #1: No

Reviewer #2: No

4. Is the manuscript presented in an intelligible fashion and written in standard English?

Reviewer #1: Yes

Reviewer #2: Yes

5. Review Comments to the Author

Reviewer #1: I have no major concern. I only have 2 minor issues-

1) In the whole exome sequencing studies, please outline the filtering steps undertaken to reach at 67 coding variants.

2) please make all data publicly available and not by request to author. Please upload mutation data to dbSNP.

Reviewer #2: In this manuscript, the authors found that the mutations in the MITF gene and C2orf74 are potentially the causal factor for a family with multiple individuals affected by Waardenburg syndrome type II. The potential mutants were first identified in two affected individuals with whole-exome sequencing (WES) and then verified in other members with Sanger sequencing. Based on the genotypes of family members, MITF and C2orf74 are highly likely the genes leading to this disease in this family.

My first concern is that not enough details were provided in the genetic analysis. More details need to be provided instead of simply saying “Various filters were applied and coding variants were reduced to 67”. Otherwise, the chosen mutation can be an artifact that happens to segregate well. Additionally, the WES data or genotype data is not available for validation by other researchers. Finally, there are quite many grammar errors that need to be fixed (see below for some examples).

Page 9: change “Further subtypes do exists” to “Further subtypes do exist”.

Page 9: change “an individual patients” to “an individual patient”.

Page 9: change “mutations in PAX3 gene is” to “mutation of PAX3 gene is”.

Page 9: change “Majority of” to “The majority of”.

Page 10: change “20-30% patients” to “20-30% of patients”.

Page 11: change “The read length of the paired end” to “The length of the paired end read”.

Page 11: “Reads were aligned to the human reference genome (GRCh38/hg38) and variants were annotated”. How variants were called?

Page 11: change “variant specific” to “variant-specific”.

Page 11: change “Detailed examination rule out presence” to “Detailed examination ruled out the presence”

Page 12: change “hypothesis free” to “hypothesis-free”.

Page 12: change “the variants in MITF and C2orf74 segregates” to “the variants in MITF and C2orf74 segregate”.

Page 12: change “Sanger sequencing validate” to “Sanger sequencing validates”.

Page 13: change “It is highly heterogeneous syndrome” to “It is a highly heterogeneous syndrome”

Page 14: references were needed for “Mutations in melanocyte inducing transcription factor (MITF), coding for a basic helix-loop-helix (BHLH) leucine zipper protein, are known to cause the WS2 phenotype”.

6. PLOS authors have the option to publish the peer review history of their article (what does this mean?). If published, this will include your full peer review and any attached files.

Reviewer #1: No

Reviewer #2: No

---

## [Author Response · Author response to Decision Letter 0]

27 Dec 2020

PONE-D-20-28113

Frameshift variant in MITF gene in a large family with Waardenburg Syndrome type II and a Co-segregation of a C2orf74 variant

PLOS ONE

Dear Editor

I am very thankful to you for sending the manuscript to reviewers for evaluation. We have revised the manuscript in view of reviewer’s comments/suggestions. Moreover, manuscript has been formatted according to the PLOS ONE style. Furthermore, raw data has been deposited to the public repositories. Changes made in the manuscript has been highlighted. Please find below point-by-point response to the comments. 

 Editorial Comments:

Comment

Response:

Manuscript has been formatted according to the PLOS ONE style. 

Comment

Response:

Raw reads (fastq files) have been uploaded to sequence read archives 

(SRA) of NCBI. The project details can be accessed using the BioProject accession number PRJNA687138. Link for data accession is https://www.ncbi.nlm.nih.gov/Traces/study/?acc=PRJNA687138

Comment

Response:

Moved to Methods section

Reviewers' comments:

Reviewer's Responses to Questions

Comments to the Author

1. Is the manuscript technically sound, and do the data support the conclusions?

Reviewer #1: Yes

Reviewer #2: Partly

2. Has the statistical analysis been performed appropriately and rigorously?

Reviewer #1: Yes

Reviewer #2: Yes

3. Have the authors made all data underlying the findings in their manuscript fully available?

Reviewer #1: No

Reviewer #2: No

4. Is the manuscript presented in an intelligible fashion and written in standard English?

Reviewer #1: Yes

Reviewer #2: Yes

5. Review Comments to the Author

Comment

Reviewer #1: I have no major concern. I only have 2 minor issues-

1) In the whole exome sequencing studies, please outline the filtering steps undertaken to reach at 67 coding variants.

Response:

Following statement has been added to the text. Please see page 6. 

Various filter were applied to search for potential candidate variants including depth of coverage (DP >10), genotype quality (GQ >20), and unreported SNP. Only variants with low allelic frequency (below 1 % in 1000G, ExAC) were taken into consideration. These settings yielded 67 rare coding variants.

Comment

2) please make all data publicly available and not by request to author. Please upload mutation data to dbSNP.

Response:

Raw reads (fastq files) have been uploaded to sequence read archives 

(SRA) of NCBI. The project details can be accessed using the BioProject accession number PRJNA687138. Link for data accession is https://www.ncbi.nlm.nih.gov/Traces/study/?acc=PRJNA687138

Reviewer #2: In this manuscript, the authors found that the mutations in the MITF gene and C2orf74 are potentially the causal factor for a family with multiple individuals affected by Waardenburg syndrome type II. The potential mutants were first identified in two affected individuals with whole-exome sequencing (WES) and then verified in other members with Sanger sequencing. Based on the genotypes of family members, MITF and C2orf74 are highly likely the genes leading to this disease in this family.

Comment: My first concern is that not enough details were provided in the genetic analysis. More details need to be provided instead of simply saying “Various filters were applied and coding variants were reduced to 67”. Otherwise, the chosen mutation can be an artifact that happens to segregate well. Additionally, the WES data or genotype data is not available for validation by other researchers. Finally, there are quite many grammar errors that need to be fixed (see below for some examples).

Response: Various filter were applied to search for potential candidate variants including depth of coverage (DP >10), genotype quality (GQ >20), and unreported SNP. Only variants with low allelic frequency (below 1 % in 1000G, ExAC) were taken into consideration. These settings yielded 67 rare coding variants.

Comment: Page 9: change “Further subtypes do exists” to “Further subtypes do exist”.

Response: Fixed

Comment: Page 9: change “an individual patients” to “an individual patient”.

Response: Fixed

Comment: Page 9: change “mutations in PAX3 gene is” to “mutation of PAX3 gene is”.

Response: Done

Comment: Page 9: change “Majority of” to “The majority of”.

Response: Done

Comment: Page 10: change “20-30% patients” to “20-30% of patients”.

Response: Done

Comment: Page 11: change “The read length of the paired end” to “The length of the paired end read”.

Response: Done

Comment: Page 11: “Reads were aligned to the human reference genome (GRCh38/hg38) and variants were annotated”. How variants were called?

Response: Genome Analysis Toolkit (GATK) was used to call the variants.

Comment: Page 11: change “variant specific” to “variant-specific”.

Response: Done

Comment: Page 11: change “Detailed examination rule out presence” to “Detailed examination ruled out the presence”

Response: Done

Comment: Page 12: change “hypothesis free” to “hypothesis-free”.

Response: Done

Comment: Page 12: change “the variants in MITF and C2orf74 segregates” to “the variants in MITF and C2orf74 segregate”.

Response: Done

Comment: Page 12: change “Sanger sequencing validate” to “Sanger sequencing validates”.

Response: Done

Comment: Page 13: change “It is highly heterogeneous syndrome” to “It is a highly heterogeneous syndrome”

Response: Done

Comment: Page 14: references were needed for “Mutations in melanocyte inducing transcription factor (MITF), coding for a basic helix-loop-helix (BHLH) leucine zipper protein, are known to cause the WS2 phenotype”.

Response: References added 

6. PLOS authors have the option to publish the peer review history of their article (what does this mean?). If published, this will include your full peer review and any attached files.

Do you want your identity to be public for this peer review? For information about this choice, including consent withdrawal, please see our Privacy Policy.

Reviewer #1: No

Reviewer #2: No

---

## [Decision Letter · Decision Letter 1]

11 Jan 2021

PONE-D-20-28113R1

Frameshift variant in MITF gene in a large family with Waardenburg Syndrome type II and a Co-segregation of a C2orf74 variant

PLOS ONE

Dear Dr. Basit,

Thank you for submitting your manuscript to PLOS ONE. After careful consideration, we feel that it has merit but does not fully meet PLOS ONE’s publication criteria as it currently stands. Therefore, we invite you to submit a revised version of the manuscript that addresses the points raised during the review process.

We look forward to receiving your revised manuscript.

Kind regards,

Obul Reddy Bandapalli, MSc, PhD

Academic Editor

PLOS ONE

Reviewers' comments:

Reviewer's Responses to Questions

**Comments to the Author**

1. If the authors have adequately addressed your comments raised in a previous round of review and you feel that this manuscript is now acceptable for publication, you may indicate that here to bypass the “Comments to the Author” section, enter your conflict of interest statement in the “Confidential to Editor” section, and submit your "Accept" recommendation.

Reviewer #1: All comments have been addressed

Reviewer #2: All comments have been addressed

2. Is the manuscript technically sound, and do the data support the conclusions?

Reviewer #1: Yes

Reviewer #2: Yes

3. Has the statistical analysis been performed appropriately and rigorously? 

Reviewer #1: Yes

Reviewer #2: Yes

4. Have the authors made all data underlying the findings in their manuscript fully available?

Reviewer #1: Yes

Reviewer #2: Yes

5. Is the manuscript presented in an intelligible fashion and written in standard English?

Reviewer #1: Yes

Reviewer #2: Yes

6. Review Comments to the Author

Reviewer #1: (No Response)

Reviewer #2: The authors have addressed my concerns. Just some suggestions: add software version for GATK and others. For gnomAD and other databases, the version is also necessary. Also, it will be recommended to include a table to summarize the 67 variants (or just some more likely variants), including their location in genome, allele frequency in gnomAD, gene, protein change, mutation affect prediction, number of individuals with the variant, and other important information that showing that they are risk mutations.

7. PLOS authors have the option to publish the peer review history of their article (what does this mean?). If published, this will include your full peer review and any attached files.

Reviewer #1: No

Reviewer #2: No

---

## [Author Response · Author response to Decision Letter 1]

21 Jan 2021

Dear Editor

I am very thankful to you for providing me the comments of a reviewer on the revised version of the manuscript. We have revised the manuscript in view of reviewer’s comments/suggestions. Please find below response to the comments of a reviewer. 

Reviewer #1: (No Response)

Reviewer #2: The authors have addressed my concerns. Just some suggestions: add software version for GATK and others. For gnomAD and other databases, the version is also necessary. Also, it will be recommended to include a table to summarize the 67 variants (or just some more likely variants), including their location in genome, allele frequency in gnomAD, gene, protein change, mutation affect prediction, number of individuals with the variant, and other important information that showing that they are risk mutations.

Response

Version numbers for GATK, dbSNP, and GnomAD have been added. Moreover a table containing information of all 67 variants have been added as a supplementary data. Please see supplementary table 1.

---

## [Editor Report · Decision Letter 2]

22 Jan 2021

Frameshift variant in MITF gene in a large family with Waardenburg Syndrome type II and a Co-segregation of a C2orf74 variant

PONE-D-20-28113R2

Dear Dr. Basit,

We’re pleased to inform you that your manuscript has been judged scientifically suitable for publication and will be formally accepted for publication once it meets all outstanding technical requirements.

Kind regards,

Obul Reddy Bandapalli, MSc, PhD

Academic Editor

PLOS ONE
---

## [Editor Report · Acceptance letter]

28 Jan 2021

PONE-D-20-28113R2 

Frameshift variant in *MITF* gene in a large family with Waardenburg Syndrome type II and a Co-segregation of a *C2orf74* variant 

Dear Dr. Basit:

I'm pleased to inform you that your manuscript has been deemed suitable for publication in PLOS ONE. Congratulations! Your manuscript is now with our production department. 

Kind regards, 

on behalf of

Dr. Obul Reddy Bandapalli 

Academic Editor

PLOS ONE